# Long-Term Ringing Data on Migrating Passerines Reveal Overall Avian Decline in Europe

**Tina Petras** [1,2] **and Al Vrezec** [3,4,*]

1   Institute of the Republic of Slovenia for Nature Conservation, Trg Etbina Kristana 1, SI-6310 Izola, Slovenia
2   Biotechnical Faculty, Department of Biology, University of Ljubljana, Jamnikarjeva 101,
    SI-1000 Ljubljana, Slovenia
3   National Institute of Biology, Večna pot 111, SI-1000 Ljubljana, Slovenia
4   Slovenian Museum of Natural History, Prešernova 20, SI-1000 Ljubljana, Slovenia
*   Correspondence: al.vrezec@nib.si

**Abstract:** The loss of biodiversity is shaping today's environment. Bird ringing is a citizen science research tool that can determine species population dynamics and trends over a large geographic area. We used a 17-year time series to assess population trends of 74 passerine species based on ringing data from autumn migration in Slovenia (south-central Europe). We defined seven guilds of species according to geographic location, ecological, migratory, breeding, and life-history traits. Almost all guilds showed declining trends, except for the group of species of northeastern European origin, which showed a stable trend. The greatest decline was in low-productivity wetland specialists. Forest birds, seed-eaters, and high-productivity species experienced the smallest declines. The general declines in avifauna across a range of life-history and behavioural traits, and across a range of spatial and ecological scales, suggest widespread environmental change in Europe. Our data indicates that recent trends are toward ecosystem homogeneity, with an impoverished avifauna, including a few species that are increasing in abundance. These are the species with higher productivity and flexible behaviour, such as short-distance migrants, that have the greatest chance of prevailing in the recently rapidly changing environment because of their ability to adapt to changes in a timely manner.

**Keywords:** bird population trends; ecological traits; life-history traits; migratory traits; recovery analysis; avifaunal change

## 1. Introduction

Biodiversity loss is one of the most alarming processes triggered by human-induced environmental and climate changes [1–3]. Despite all efforts, reversing the decline still remains largely unrealized [3–5]. The basis for effective management and conservation is reliable data. This is where monitoring plays an essential role, as it allows us to determine the status of species–the distribution and direction of population changes–and to identify the causes of these changes [6]. The abundance and distribution of species are influenced by ecological and environmental processes that occur at local, regional, landscape, and continental scales [7], so there is a high need for monitoring at multiple scales [8,9].

Large-scale and long-term monitoring mainly involves volunteers in citizen science projects that can provide larger data sets. However, such data are usually subject to more biases and often lead to a lack of statistical power and misinterpretation of the data [10,11]. Here, the large sample size of robust data could mask biases [12–14], and furthermore, more sophisticated methods of data analysis could increase statistical power in trend estimates and species distribution [15,16]. Because data collected at multiple locations and spatial scales are essential to detect variation in factors affecting species throughout their range [17,18], many citizen science projects have expanded in recent decades, such as eBird, Big Garden Birdwatch, and North American Breeding Bird Survey [19,20]. However, species can already exhibit highly variable regional population trends at the continental

scale [21], which may depend on differences in habitat quality, productivity, survival, as well as population size, connectivity, and losses during migration [22–26].

Bird ringing is a research tool for determining species population dynamics and trends over a larger geographic scale, conducted primarily by trained and experienced volunteers [27–34]. Ringing can provide additional support for cross-checking trend estimates obtained by monitoring breeding bird counts when the population of origin is considered. During bird migration, species congregate at stopover sites where ringing stations are placed, primarily to monitor autumn bird migration dynamics. Such ringing of migratory birds provides long continuous records for 100 years or more, and can provide valuable insights into the long-term population dynamics of target species [35]. If we know the origin of species' breeding ranges, we can relate bird population dynamics to factors that are likely to influence them. Species population dynamics and trends may vary within their range due to (1) different environmental conditions [36], (2) marginality of populations [37,38], (3) different socioeconomic status [39], and/or (4) different conservation efforts in different countries [7]. Trend analyses allow us to detect population changes; but, without identifying the processes that caused them, we cannot understand and eliminate the factors affecting populations [40]. Therefore, trends need to be linked to ecological factors and species life history to gain a better understanding of the factors that influence population dynamics [28,41].

Here, we aimed to compare how well population trend estimates of migratory passerine bird species based on ringing data from autumn migration are compatible with population trends estimated from surveys of breeding birds. We were also interested in the group of traits by which population trends in migratory bird avifauna are most strongly influenced. Because ringing data were continuously collected but not systematically collected, data analysis and interpretation are particularly important to avoid the risk of misleading conclusions and to draw more reliable inferences [17]. The objectives of the study were to (i) estimate population trends of passerine species during autumn migration over a 17-year period based on ringing data in Slovenia [32]; (ii) analyse recoveries to determine breeding origins of the species studied; (iii) investigate whether guilds of species with similar movement status, life history traits, common ecological characteristics, and geographic origin share a common trend direction.

## 2. Materials and Methods

### 2.1. Study Area

Slovenia is an important passage area for migratory birds due to its location at the junction between Central Europe and the Balkans (Figure 1) [42]. Western Palearctic passerine birds migrate mainly along four migration routes: south-eastern (Balkans), central (Apennines), western (Atlantic), and eastern (India), with the first three routes passing through Slovenia [43]. Along the south-eastern European migration route, populations from Central Asia, Russia, and northern, north-western, and central Europe migrate and continue their journey across the Balkan and Apennine Peninsulas, and further along the Adriatic Sea or across the Bosporus to the eastern parts of Africa [43]. Populations from the northern regions of Russia and northern and central Europe migrate along the central migration route, mostly via the Balkan and Apennine Peninsulas, to the western and central regions of Africa. The western migration route, used mainly by northern Russian and northern European populations, includes the regions above the Apennine and Balkan Peninsulas, and leads westward across Gibraltar to West Africa [43]. Although this route is mainly north of the Alps, some birds from Slovenia and central Europe use the southern part of the western migration route to avoid crossing the Alps, the central Mediterranean, and the central Sahara Desert [44].

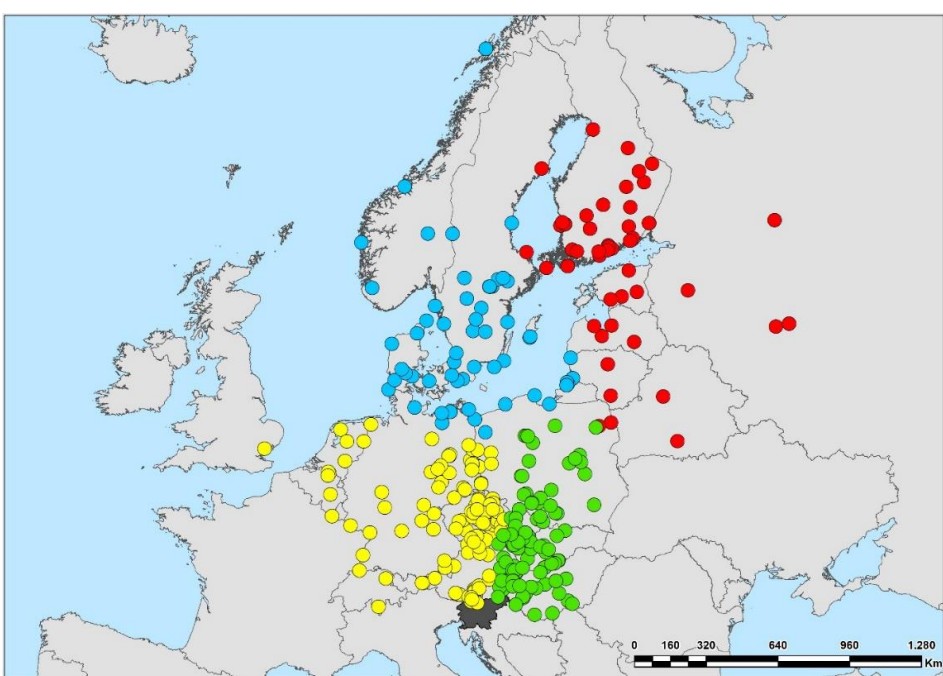

**Figure 1.** Spatial clustering of migratory passerines according to recoveries with defining four clusters of bird origin (yellow dots—North Western (NW), green dots—North Eastern (NW), red dots—North-North-Eastern (NNE), blue dots—North-Northern (NN)) according to the position to Slovenia (shaded in dark) within Europe.

### 2.2. Bird-Ringing Database and Species Selection

We used ringing data of passerine birds from the Slovenian Bird Ringing Scheme organised by the Slovenian Museum of Natural History. The non-standardised method of ringing birds involves several biases, so we examined the variability in the number of ringing sites and the variability of ringed individuals among years to account for the heterogeneity of the data; see details in [31]. The coefficients of variation of the aforementioned variables and the average number of annually ringed birds with standard deviation were estimated for each species, and in addition, the maximum number of sites and the number of years species were ringed during the 17-year period for a given species was reported (Supplementary Materials, Table S1). Trends were estimated during the period between 2000 and 2016 for the autumn migration period, from July 15 to December 31. During the 17-year period, ringing stations were operated more or less regularly in the autumn, with mist netting as the predominant method [45]. Breeding and migratory status were recorded for each species (Supplementary Materials, Table S2). Only species ringed in more than eight years (more than 50% of the study period) and with an average of at least five ringed individuals per year were included in trend modelling and other statistical analyses. With this arbitrary data selection, we attempted to eliminate biases due to a small population size that could lead to low statistical power and associated error II [46–48]. Species whose taxonomic status was unclear in the past, such as the Common/Lesser Redpoll (*Acanthis flammea/cabaret*) [49], were excluded from the analysis due to uncertainty in identification.

### 2.3. Modelling Population Dynamics and Trends

For modelling migratory bird population trends, we applied the universal model that best fit the broadest range of species, according to Petras and Vrezec [32]. We fitted our data using a generalised additive model (GAM) with thin plate spline smoothing, which is particularly useful for noisy data [50,51] (package gam) [52]. The fitted values obtained with GAM were used for trend estimates in the generalised linear model (GLM), which allowed us to build a linear relationship between the response and predictor variables, even if their

relationship was not linear [53]. Poisson regression was used here because the response variable had a Poisson distribution. For species with rejected models (nonsignificant GAM models and/or models with ≤ 10% explained deviance) that had a higher coefficient of variation among years (≥ 30%), we performed a 3-year moving average of the number of individuals (Supplementary Materials, Table S1), the smooth technique commonly used in time series analyses. It reduces the noise of random or irregular fluctuations inherent in all time series and highlights long-term trends or cycles [54].

### 2.4. Recovery Analysis

To determine to which breeding populations our studied migratory bird species belonged, we analysed recovery data from the Slovenian Bird Ringing database. The bias of such data [55,56], as well as rare and scattered recoveries [57–59], may hamper such analyses. The probability of recovering a ringed bird depends on several factors that spatially and temporally vary [56,59–62]. One of the major difficulties in using ring recoveries to link migratory and breeding populations is a small representative sample to capture heterogeneity in species distribution [56]. We used the recovery data to make inferences about the breeding range of the species. All ringed pullus and individuals ringed or found north of Slovenia during the breeding season, and ringed or recovered in Slovenia during autumn migration, were included in the recovery analysis. We provided the breeding season for each species [63,64] in order to more accurately and reliably determine breeding populations based on recoveries. We clustered geographic coordinates of recoveries using the KMedians method and Manhattan distance. This method is recommended for spatial clustering because it reduces the number of obstacles and filters out obstacles that have no effect on the similarity between spatial objects [65,66]. We defined four geographic clusters north of Slovenia, reflecting the direction of bird migration to Slovenia. We then aggregated species recoveries according to their proportion in each geographic cluster, resulting from the spatial clustering. In this way, we created clusters (dendrograms) of species according to their breeding range. We used the clustering method ward.D2 and Euclidean distance (package cluster) [67].

### 2.5. Species Traits and Their Association to Population Trends

To determine which factors most influenced population trends, we clustered species by the following attributes: geographic origin, migratory behaviour (migration distance and migration status), ecological preferences (habitat and diet), and life history traits (productivity and distribution range) (Table 1 and Supplementary Materials, Table S2). Data from the literature were used to group species by the appropriate traits (Table 1), and recovery analysis was used for geographic grouping. To determine total productivity per year, we multiplied the average annual number of eggs produced per pair by the average number of nests per year. For trend analysis, we considered only the extreme values of the productivity trait: 3.0–6.9 for low productivity and 10.0–20.9 for high productivity. Because absolute numbers of birds captured widely varied among species, we converted population indices to the same decimal basis before assessing common trends for each guild. Otherwise, the indices of species that were captured in large numbers would predominate over other species and potentially bias the trend estimates for individual guilds. Indices for each trait group were calculated as the average number of species indices belonging to a given group. Trends were modelled by GAM, while trend direction was assessed by GLM. Guilds with rejected models (nonsignificant GAM models and/or models with ≤10% explained deviation by 3-year moving average) were classified as uncertain. Adjusted values of accepted models were modelled using GLM to identify trends; where a non-significant slope indicates a stable trend, while a significant slope indicates an increasing or decreasing trend. We applied Ordinal Logistic Regression (OLR; package MASS) [68] to examine which species traits had the greatest influence on the direction of the population trend. This regression modelled the relationship between trend (response variable) as an ordered categorical variable and traits (explanatory variables) [69].

**Table 1.** Abbreviations of traits with given species number in each trait group, statistical type of data and data source.

| Species' Trait Category and Trait Type | Abbreviation of Categories | Definition | Number of Species | Type of Statistical Data | Data Source |
|---|---|---|---|---|---|
| **ECOLOGICAL TRAITS** | | | | | |
| Cultural landscape | CL | Species living in cultural landscape | 27 | Categorical nominal | [63,64] |
| Forest | FOR | Species living in forest | 36 | | |
| Wetland | WET | Wetland specialists | 11 | | |
| Diet | SEEDS | Seed-eating birds | 13 | | |
| | INVER | Species feeding on insects and other invertebrates | 61 | | |
| **MIGRATORY TRAITS, BREEDING AND ABUNDANCE STATUS** | | | | | |
| Migratory status in Europe | PAS | Passenger, including accidental passenger | 33 | Categorical nominal | [64,70–72] |
| | RES | Resident in Europe (in Slovenia: resident; resident and passenger; wintering species) | 31 | | |
| | NOM | Nomadic and irruptive species | 10 | | |
| Migration distance | LM | Long-distance migrant: Sub-Saharan-Africa and Asia | 30 | | |
| | SM | Short-distance migrant | 35 | | |
| | NM | Non-migrant species | 9 | | |
| Breeding status in Slovenia | RB | Regular breeder | 68 | | |
| | OB | Occasional breeder | 4 | | |
| | NB | Non-breeding species | 2 | | |
| Breeding distribution range in Europe | CBE | Common breeder and widespread breeding species in most of Europe | 56 | | |
| | SBE | Species with scattered or restricted distribution to few parts of Europe | 18 | | |
| **LIFE-HISTORY TRAITS** | | | | | |
| Reproduction | | avg. no. of annually produced eggs per pair * avg. no. of annual nests | | Numerical and categorical ordinal | [61,73–79] |
| | MP | Low productivity: 3.0–6.9 | 22 | | |
| | LP | High productivity: 10.0–20.9 | 26 | | |
| **GEOGRAPHICAL ORIGIN** | | | | | |
| Geographical origin (recoveries according to position of Slovenia) | NE | North-Eastern | 11 (122 recoveries) | Categorical nominal | Analyses in this article |
| | NW | North-Western | 14 (149 recoveries) | | |
| | NN | North-Northern | 8 (80 recoveries) | | |
| | NNE | North-North-Eastern | 6 (54 recoveries) | | |
| Latitude (N) Longitude (E) Migratory distance (recoveries) | | | | Numerical | Slovenian bird ringing database |

The statistical program GeoDa 1.20 [80] was used for spatial hierarchical clustering of geographical coordinates for recoveries. All other statistical analyses were made in R version 4.1.3 [81].

## 3. Results

### 3.1. Species Selection

During the study period from 2000 to 2016, 120 passerine species were ringed in Slovenia during autumn migration (Supplementary Materials, Table S1). Of these, 46 species were excluded from further analyses because the number of years in which

the species were ringed was too low (i.e., species ringed in <8 years or on average <5 individuals ringed per year). A total of 74 species were included in the modelling analyses. Only seven of the modelled species, before conducting the smoothing procedure, had more homogeneous data with respect to variation among individuals, with a corresponding coefficient < 30% (Supplementary Materials, Table S1).

### 3.2. Population Dynamics and Trends

Population dynamics modelling was performed for the 69 species using accepted models, i.e., significant models and/or models with ≥10% explained deviance (Table 2; Supplementary Materials, Figure S1). Of the accepted models, only the models for Hawfinch (*Coccothraustes coccothraustes*) and Eurasian Treecreeper (*Certhia familiaris*) reflected a higher degree of uncertainty with <20% explained deviance. In most cases (44), the model fitted the data very well (>50% explained deviance), and 48 models were significant. The majority of species (43 or 58%) had declining population trends, 20 species had increasing population trends, six species had stable populations, and five species had rejected models (Table 2). Our results agreed well with reference trend estimates from breeding bird surveys across Europe [78,82], and to some extent from Slovenian breeding bird monitoring [83]. The largest deviations from European or Slovenian breeding bird trends were found in only four species: Short-toed Treecreeper (*Certhia brachydactyla*), Brambling (*Fringilla montifringilla*), House Sparrow (*Passer domesticus*), and Common Starling (*Sturnus vulgaris*).

### 3.3. Species Guilds

We analysed 405 recoveries for 39 species. After spatially clustering the latitudinal and longitudinal coordinates of the recoveries, we identified four guilds indicating the directions from which the birds reached Slovenia (Figure 1 and Supplementary Materials, Table S3). In addition, a dendrogram was used to cluster species into four guilds according to their breeding range (Figure 2). The results of spatial clustering analyses and hierarchical clustering analysis, in which we preliminarily determined four guilds, showed that most of the recoveries and species came from NW (149 recoveries and 14 species) and NE (122 recoveries and 11 species) directions according to the position of Slovenia (Figures 1 and 2). The number of recoveries decreased towards the north and north-east. There were 80 recoveries for eight species from NN, and 54 records for six species from NNE. Siskin (*Spinus spinus*) was the most heterogeneous, with 22 records from 11 states. The Common Chiffchaff (*Phylloscopus collybita*), on the other hand, was recorded only seven times in a single state (Czech Republic). Sedge Warbler (*Acrocephalus schoenobaenus*), Blackcap (*Sylvia atricapilla*), and Dunnock (*Prunella modularis*) were the species with the most recoveries, covering a reasonable portion of their European breeding range. In total, we defined seven species guilds according to geographic location, ecological preferences, migratory and breeding behaviour, and life-history traits (Supplementary Materials, Table S2).

### 3.4. Population Trends within Species Guilds

Almost all guilds showed declining trends, except for the species guild with NE geographic origin, which showed a stable trend (Table 3 and Supplementary Materials, Figures S2–S4). Declines were greatest for wetland specialists, low-productivity species, cultural landscape species, invertebrate foragers, long-distance migrants, and non-migrant species. In contrast, the rate of decline was lowest for granivorous and forest-dwelling species, short-distance migrants, nomads, and high-productivity species (Table 3).

**Table 2.** General additive models (GAM) of population dynamics for the 74 passerine species. Estimated annual growth rate (trend) was assessed in a GLM. Reference data for trend-matching were [78,82,83]. Significant codes: *** $p < 0.001$, ** $p < 0.01$, * $p < 0.05$. Abbreviation: DE: deviance explained, SE: standard error.

| Species | Estimate (GAM) ± SE | F | DE (%) | Model Acceptance | Estimated Annual Growth Rate (GLM) | Trend | Trend-Match with Ref. Data |
|---|---|---|---|---|---|---|---|
| *Acrocephalus arundinaceus* | 0.698 ± 0.030 | 1.9 | 20.2 | yes | +0.013 *** | Increase | yes |
| *Acrocephalus melanopogon* | 0.054 ± 0.009 | 4.0 * | 50.7 | yes | −0.089 *** | Decline | yes |
| *Acrocephalus palustris* | 2.840 ± 0.095 | 16.0 *** | 85.7 | yes | −0.051 *** | Decline | yes |
| *Acrocephalus schoenobaenus* | 6.224 ± 0.219 | 2.2 | 67.3 | yes | +0.000 | Stable | yes |
| *Acrocephalus scirpaceus* | 7.583 ± 0.347 | 8.6 * | 36.5 | yes | −0.027 *** | Decline | yes |
| *Aegithalos caudatus* | 0.606 ± 0.031 | 3.3 * | 66.2 | yes | −0.038 *** | Decline | yes |
| *Anthus trivialis* | 0.175 ± 0.005 | 10.8 ** | 45.4 | yes | −0.023 *** | Decline | yes |
| *Carduelis carduelis* | 0.569 ± 0.054 | 3.3 * | 47.7 | yes | −0.023 *** | Decline | yes |
| *Certhia brachydactyla* | 0.022 ± 0.002 | 6.2 * | 29.5 | yes | −0.059 *** | Decline | no |
| *Certhia familiaris* | 0.041 ± 0.002 | 2.6 | 17.1 | yes | +0.020 *** | Increase | unknown |
| *Cettia cetti* | 0.047 ± 0.003 | 5.9 ** | 81.6 | yes | −0.061 *** | Decline | yes |
| *Chloris chloris* | 1.110 ± 0.034 | 5.8 * | 85.2 | yes | −0.011 * | Decline | yes |
| *Coccothraustes coccothraustes* | 0.108 ± 0.011 | 2.2 | 13.3 | yes | +0.032 *** | Increase | yes |
| *Curruca communis* | 0.929 ± 0.036 | 17.1 *** | 80.4 | yes | −0.049 *** | Decline | yes |
| *Curruca curruca* | 0.545 ± 0.018 | 3.4 * | 70.4 | yes | −0.023 *** | Decline | yes |
| *Curruca nisoria* | 0.045 ± 0.003 | 2.7 | 63.7 | yes | −0.039 *** | Decline | yes |
| *Cyanecula svecica* | 0.087 ± 0.006 | 3.2 | 46.5 | yes | +0.013 *** | Increase | yes |
| *Cyanistes caeruleus* | 2.291 ± 0.095 | 3.1 | 76.5 | yes | +0.002 | Stable | yes |
| *Delichon urbicum* | 0.044 ± 0.005 | 9.7 ** | 92.2 | yes | −0.013 *** | Decline | yes |
| *Emberiza cia* | 0.034 ± 0.003 | 29.5 *** | 66.3 | yes | −0.116 *** | Decline | yes |
| *Emberiza cirlus* | 0.027 ± 0.001 | 4.9 * | 76.9 | yes | +0.040 *** | Increase | yes |
| *Emberiza citrinella* | 0.197 ± 0.008 | 0.1 | 0.8 | no | - | Uncertain | unknown |
| *Emberiza schoeniclus* | 1.314 ± 0.113 | 8.5 * | 36.4 | yes | −0.052 *** | Decline | yes |
| *Erithacus rubecula* | 9.227 ± 0.340 | 4.5 * | 50.3 | yes | −0.023 *** | Decline | yes |
| *Ficedula hypoleuca* | 0.216 ± 0.015 | 3.5 * | 66.3 | yes | −0.022 *** | Decline | yes |
| *Fringilla coelebs* | 0.907 ± 0.059 | 5.8 ** | 58.7 | yes | −0.054 *** | Decline | yes |
| *Fringilla montifringilla* | 0.222 ± 0.016 | 2.6 | 46.0 | yes | +0.026 *** | Increase | no |
| *Garrulus glandarius* | 0.027 ± 0.001 | 4.9 * | 84.7 | yes | −0.018 *** | Decline | yes |
| *Hippolais icterina* | 0.668 ± 0.062 | 8.1 ** | 47.5 | yes | −0.067 *** | Decline | yes |
| *Hippolais polyglotta* | 0.017 ± 0.002 | 2.2 | 39.9 | yes | +0.057 *** | Increase | yes |
| *Hirundo rustica* | 16.715 ± 2.785 | 1.2 | 39.8 | yes | −0.048 *** | Decline | yes |
| *Lanius collurio* | 0.367 ± 0.018 | 7.9 ** | 74.7 | yes | −0.052 *** | Decline | yes |
| *Linaria cannabina* | 0.074 ± 0.009 | 2.2 | 61.5 | yes | −0.007 | Stable | yes |
| *Locustella fluviatilis* | 0.033 ± 0.003 | 7.4 ** | 54.9 | yes | −0.081 *** | Decline | yes |
| *Locustella luscinioides* | 0.080 ± 0.006 | 3.5 * | 48.5 | yes | −0.022 *** | Decline | unknown |
| *Locustella naevia* | 0.301 ± 0.020 | 8.9 ** | 56.5 | yes | −0.058 *** | Decline | yes |
| *Lophophanes cristatus* | 0.126 ± 0.005 | 7.1 ** | 90.5 | yes | +0.048 *** | Increase | unknown |
| *Loxia curvirostra* | 0.022 ± 0.002 | 3.4 * | 74.1 | yes | +0.043 *** | Increase | yes |
| *Luscinia luscinia* | 0.051 ± 0.003 | 2.3 | 35.5 | yes | −0.031 *** | Decline | yes |
| *Luscinia megarhynchos* | 0.351 ± 0.013 | 4.9 * | 64.6 | yes | +0.028 *** | Increase | yes |
| *Motacilla alba* | 0.036 ± 0.009 | 1.3 | 38.1 | yes | −0.015 *** | Decline | yes |
| *Motacilla cinerea* | 0.013 ± 0.001 | 3.7 * | 48.1 | yes | +0.048 *** | Increase | yes |
| *Motacilla flava* | 0.087 ± 0.017 | 3.5 * | 62.6 | yes | +0.055 *** | Increase | yes |
| *Muscicapa striata* | 0.145 ± 0.007 | 6.5 ** | 70.2 | yes | −0.023 *** | Decline | yes |
| *Oriolus oriolus* | 0.013 ± 0.000 | 3.1 | 65.7 | yes | −0.026 *** | Decline | yes |
| *Parus major* | 4.007 ± 0.239 | 0.4 | 3.5 | no | - | Uncertain | unknown |
| *Passer domesticus* | 0.458 ± 0.024 | 4.4 * | 63.1 | yes | −0.040 *** | Decline | no |
| *Passer italiae* | 0.022 ± 0.002 | 11.2 ** | 91.1 | yes | +0.093 *** | Increase | unknown |
| *Passer montanus* | 2.248 ± 0.161 | 3.7 * | 45.8 | yes | −0.033 *** | Decline | yes |
| *Periparus ater* | 1.929 ± 0.143 | 3.2 | 59.8 | yes | −0.009 *** | Decline | yes |
| *Phoenicurus ochruros* | 0.067 ± 0.002 | 4.0 * | 72.5 | yes | +0.018 *** | Increase | yes |
| *Phoenicurus phoenicurus* | 0.112 ± 0.005 | 3.6 * | 79.4 | yes | +0.024 *** | Increase | yes |
| *Phylloscopus collybita* | 3.911 ± 0.156 | 3.7 * | 55.3 | yes | +0.009 *** | Increase | yes |
| *Phylloscopus sibilatrix* | 0.993 ± 0.116 | 1.4 | 38.7 | yes | −0.018 *** | Decline | yes |
| *Phylloscopus trochilus* | 0.507 ± 0.023 | 5.2 * | 48.2 | yes | −0.032 *** | Decline | yes |
| *Poecile montanus* | 0.079 ± 0.002 | 10.1 ** | 93.5 | yes | +0.001 | Stable | unknown |
| *Poecile palustris* | 0.233 ± 0.013 | 0.1 | 6.9 | no | - | Uncertain | unknown |
| *Prunella modularis* | 11.087 ± 0.403 | 5.2 * | 28.6 | yes | −0.019 *** | Decline | yes |
| *Pyrrhula pyrrhula* | 0.043 ± 0.001 | 4.4 * | 74.5 | yes | −0.010 *** | Decline | yes |
| *Regulus ignicapilla* | 0.070 ± 0.005 | 1.9 | 45.5 | yes | +0.012 *** | Increase | yes |
| *Regulus regulus* | 3.451 ± 0.148 | 6.4 * | 86.5 | yes | +0.003 | Stable | yes |
| *Remiz pendulinus* | 0.517 ± 0.045 | 3.8 * | 74.4 | yes | −0.038 *** | Decline | unknown |
| *Riparia riparia* | 0.185 ± 0.040 | 5.6 * | 27.2 | yes | −0.112 *** | Decline | unknown |
| *Saxicola rubetra* | 0.088 ± 0.002 | 8.6 ** | 91.5 | yes | −0.012 *** | Decline | yes |
| *Saxicola rubicola* | 0.074 ± 0.004 | 3.6 * | 78.9 | yes | −0.0541 *** | Decline | yes |
| *Serinus serinus* | 0.190 ± 0.014 | 2.2 | 51.2 | yes | −0.006 | Stable | yes |
| *Sitta europaea* | 0.054 ± 0.004 | 1.3 | 28.2 | yes | +0.027 *** | Increase | yes |
| *Spinus spinus* | 2.199 ± 0.218 | 5.5 * | 85.6 | yes | −0.046 *** | Decline | yes |
| *Sturnus vulgaris* | 0.303 ± 0.023 | 5.1 * | 85.3 | yes | +0.031 *** | Increase | no |
| *Sylvia atricapilla* | 41.656 ± 1.257 | 3.6 | 40.3 | yes | −0.016 *** | Decline | yes |
| *Sylvia borin* | 14.386 ± 0.432 | 9.1 ** | 91.0 | yes | −0.038 *** | Decline | yes |
| *Troglodytes troglodytes* | 0.217 ± 0.012 | 2.6 | 61.2 | yes | +0.010 ** | Increase | yes |
| *Turdus merula* | 1.298 ± 0.056 | 0.1 | 0.8 | no | - | Uncertain | unknown |
| *Turdus philomelos* | 0.588 ± 0.042 | 0.5 | 3.3 | no | - | Uncertain | unknown |

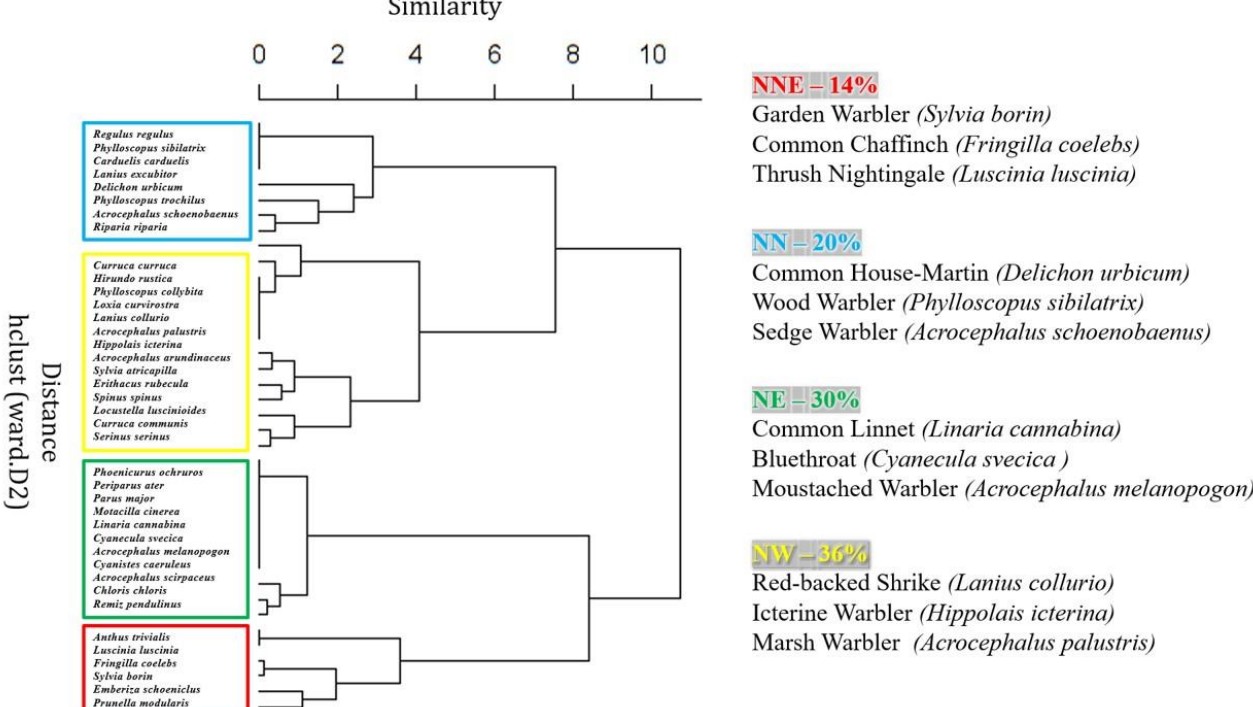

**Figure 2.** Species guilds according to four geographical clusters, directions from where the species arrive according to the position of Slovenia. For particular guilds, the percentages of recoveries of each geographical cluster are given, including corresponding species with higher percentages of presence.

**Table 3.** Population trends in species trait guilds. Abbreviations: SE: standard error. Significant codes: *** $p < 0.001$, ** $p < 0.01$, * $p < 0.05$.

| Grouping by Species' Traits | Estimate (GAM) $\pm$ SE | F | Deviance Explained (%) | Estimated Annual Growth Rate (GLM) | Trend |
|---|---|---|---|---|---|
| **Habitat** | | | | | |
| Cultural landscape | $2.453 \pm 0.059$ | 9.6 ** | 65.9 | $-0.025$ *** | Decline |
| Forest | $2.568 \pm 0.077$ | 1.6 | 11.3 | $-0.008$ * | Decline |
| Wetland | $2.504 \pm 0.071$ | 17.7 *** | 74.5 | $-0.039$ *** | Decline |
| **Diet** | | | | | |
| Seed-eaters | $2.448 \pm 0.078$ | 1.4 | 10.1 | $-0.008$ * | Decline |
| Invertebrate eaters | $2.451 \pm 0.073$ | 11.4 ** | 46.7 | $-0.023$ *** | Decline |
| **Migratory status in Europe** | | | | | |
| Passage migrants | $2.452 \pm 0.074$ | 8.1 ** | 55.6 | $-0.027$ *** | Decline |
| Residents | $2.545 \pm 0.061$ | 3.1 | 62.9 | $-0.018$ ** | Decline |
| Nomadic and irruptive species | $2.385 \pm 0.110$ | 0.4 | 11.7 | $-0.001$ *** | Decline |
| **Migration distance** | | | | | |
| Long-distance migrants | $2.525 \pm 0.076$ | 7.5 ** | 55.6 | $-0.027$ *** | Decline |
| Short-distance migrant | $2.375 \pm 0.068$ | 3.2 | 20.0 | $-0.012$ ** | Decline |
| Non-migrant species | $2.931 \pm 0.085$ | 4.5 * | 80.3 | $-0.021$ *** | Decline |
| **Breeding status in Slovenia** | | | | | |
| Regular breeder | $2.563 \pm 0.065$ | 11.7 ** | 47.5 | $-0.020$ *** | Decline |
| Occasional breeder | $1.269 \pm 0.059$ | 0.0 | 0.1 | 0.001 | Uncertain |
| Non-breeding species | $2.955 \pm 0.101$ | 3.4 * | 75.1 | $-0.016$ ** | Decline |

**Table 3.** *Cont.*

| Grouping by Species' Traits | Estimate (GAM) ± SE | F | Deviance Explained (%) | Estimated Annual Growth Rate (GLM) | Trend |
|---|---|---|---|---|---|
| **Breeding distribution range in Europe** | | | | | |
| Widespread | 2.423 ± 0.068 | 4.2 * | 44.4 | −0.019 *** | Decline |
| Small or scattered range | 2.537 ± 0.063 | 10.4 ** | 64.6 | −0.026 *** | Decline |
| **Geographic origin of breeding populations** | | | | | |
| North-Eastern | 1.756 ± 0.057 | 2.2 | 70.0 | 0.005 | Stable |
| North-Western | 2.392 ± 0.078 | 10.6 ** | 44.9 | −0.024 *** | Decline |
| North-Northern | 3.060 ± 0.118 | 3.2 | 42.2 | −0.021 *** | Decline |
| North-North-Eastern | 5.374 ± 0.191 | 5.3 * | 56.0 | −0.026 *** | Decline |
| **Productivity** | | | | | |
| Low productivity | 2.327 ± 0.073 | 13.8** | 51.6 | −0.027 *** | Decline |
| High productivity | 2.797 ± 0.075 | 4.2 | 24.9 | −0.012 *** | Decline |

According to the OLR models, productivity, migratory behaviour (migratory status in Europe), and latitude had the largest influence on trend direction (Table 4). All other interactions between traits and trends were not significant.

**Table 4.** Impacts of particular trait on population trend. Bolded are interactions with $p > 0.05$.

| Species' Trait Category or Trait Type | t | p |
|---|---|---|
| Habitat | −0.797 | 0.424 |
| Diet | −0.753 | 0.451 |
| **Migratory status in Europe** | **2.648** | **0.008** |
| Migration distance | 2.028 | 0.042 |
| Breeding status in Slovenia | 0.029 | 0.976 |
| Breeding distribution range in Europe | −0.805 | 0.420 |
| **Productivity** | **2.416** | **0.015** |
| Actual migration distance (recoveries) | 1.541 | 0.123 |
| Longitude (recoveries) | −1.839 | 0.065 |
| **Latitude (recoveries)** | **2.475** | **0.013** |
| Geographic origin of breeding populations (recoveries) | 0.999 | 0.317 |

## 4. Discussion

Application of the universal model, developed for estimating population trends based on long-term and non-standardised autumn migration ringing data [32] to estimates of population trends for several species, yielded good agreement with population trends obtained from surveys of breeding birds throughout Europe (Table 2). Using a 17-year time series of the abundance index of passerine bird populations, we showed that populations of most migratory bird species were declining (62% of 69 species with accepted trend models). Analysis of trait trends showed that almost all traits examined had declining trends, suggesting that bird populations are experiencing overall negative pressure more or less independent of their traits. This suggests that, at least on the European continent, there is a general decline in passerine avifauna that may also be affecting European avifauna at broader trait levels.

### 4.1. Population Trends of Passerines

Overall, 93% of known population trends in our study, determined from ringing data during the autumn migration season, agreed well with trends in European or Slovenian breeding populations (Table 2). Four of the species studied deviated in trend direction from reference trends. These differences largely resulted from consideration of various aspects of the populations and monitoring methodology. First, different portions of bird populations could be considered when monitoring populations during the breeding season and during autumn migration. Breeding populations include the number of breeding or territorial adult birds, excluding young and floaters, whereas autumn migration popula-

tions include all individuals. Therefore, it would be possible that abundance changes in populations would be detected earlier in migratory populations because they contain an additional segment of productivity. Second, populations migrating through Slovenia do not cover entire European breeding populations. It is conceivable that populations migrating through the region are predominantly from those parts of Europe whose trends diverge from the common European breeding trend, as is the case for the Great Reed Warbler (*Acrocephalus arundinaceus*). The common European breeding trend of the Great Reed Warbler is declining [78], but most recoveries (69% from three countries) were from states with increasing populations, indicating that the geographic origin of the species yielded the trends that were consistent with our data. Finally, the differences in trend directions between the bird ringing and breeding bird survey data could be due to sampling error [46], i.e., the monitoring included a limited and unrepresentative portion of the populations. This might be reflected especially in the House Sparrow, whose Slovenian breeding population is at least stable [82,83], while the population trend according to our data and the common European data of breeding populations is decreasing [78]. Considering that the House Sparrow is resident in the whole of Europe [64], and there are no or negligible number of recoveries from abroad [84,85], the trend of the breeding population of the House Sparrow in Slovenia could probably be misjudged. Monitoring of breeding House Sparrow populations is based on Farmland Bird Index counts [83]. In farmland habitats, the population trend estimate might be biased due to the small and non-representative sample, as the predominant habitats of the species are urban areas [64].

### 4.2. Geographical Origin of Species

Recovery analysis showed that passerine populations in northern, central, and western Europe (NN, NW, NNE groups) sharply declined overall. Stable populations occurred only in the group of species that passed Slovenia from NE from eastern European countries (Table 3 and Supplementary Materials, Figure S4). These results are in contrast to Cuervo and Møller [37], who found an increase in northern populations, and are consistent with other studies that found declining populations in northern and central parts of Europe (Denmark and Germany), and mostly stable trends in eastern Europe (Czech Republic) [86]. Such differences in trend directions within some geographic groups could be due to the comparison of several countries where species differ in their population trends [18], or to the selection of species that differ in their trend directions. In our study, however, we were restricted to species for which we could demonstrate recoveries from the breeding season.

Declining common trends in almost all geographic guilds indicate unfavourable ecological conditions in much of Europe [5]. The only non-declining group, NE, showed more favourable conditions in eastern European countries with less intensive land use and greater land parcelization [39,87]. The better condition of eastern European populations compared to others has also been noted in other studies, e.g., the greater stability of forest bird populations [88] and, until the last two decades, the good condition of farmland bird populations [89]. However, the lack of a positive trend for the NE guild suggests that land use management and associated populations of bird species are following the trend direction of other European regions [89].

### 4.3. Common Trends in Trait Guilds

In particular, species with a predominant combination of guilds, including wetland specialists, farmland species, low-productivity species, non-migrants, and long-distance migrants, experienced some of the largest population declines (Table 2), such as Moustached Warbler (*Acrocephalus melanopogon*), Collared Sand Martin (*Riparia riparia*), and Cetti's Warbler (*Cettia cetti*). All of these species are wetland specialists and account for the largest proportion of species with declining trends (72%) among all trait guilds. In general, the decline of wetland specialists, with the exception of waterfowl [90,91], has been observed worldwide [92], due to the degradation of wetland habitats, which are very scattered in Europe [93]. Such habitat patterns are also reflected in the occurrence of wetland birds in

our study, with the largest proportion of species with scattered distributions found in the wetland guild (45%). Wetland management, along with the dispersed locations of wetlands, is an additional constraint for marsh passerines. These species are particularly vulnerable due to their narrow ecological niche, as they are species-specific-dependent on vegetation structure and water regime [94–96]. An increase in population size was found only in two wetland species studied, namely the Great Reed Warbler, which is widespread throughout Europe, and the Bluethroat (*Cyanecula svecica*). Population growth in wetland species has recently been only exceptionally reported [97]. These contradictory results are probably related to the selection of species, which included a considerable number of non-passerines associated with various open water habitats. Another indication of wetland specialists was the greater number of low-productivity species that predominated in this trait group (54%). However, there were no species with high productivity among the wetland specialists. Moreover, low productivity is generally associated with population declines [98,99]. In our study, the group of species with low productivity also included one of the largest proportions of declining species (39%).

During autumn migration, birds are less restricted to a particular habitat type than during spring migration [100], and many species that live in different habitats during the breeding season congregate at stopover sites [101]. Wetlands, especially marshes overgrown with reeds, are important stopover sites for many passerine birds during migration [100,102,103]. Some species, such as the European Robin (*Erithacus rubecula*), the Common Redstart (*Phoenicurus phoenicurus*), the Northern Wren (*Troglodytes troglodytes*), the Dunnock (*Prunella modularis*), the Common Chiffchaff (*Phylloscopus collybita*), and the Willow Warbler (*Phylloscopus trochilus*), which are found in scrubby habitats at the beginning of migration season, move to reedbeds at the end of migration, especially between October and November [101]. Accordingly, wetlands are also important to species that are not strictly specialised in these habitats, so destruction of these habitats affects both wetland specialists and habitat generalists.

In our study, all wetland species fed primarily on invertebrates, at least during the breeding season (Supplementary Materials, Table S2). It is conceivable that the loss of aquatic insects [104], largely caused by pesticides used to control mosquitoes [105], affects wetland bird-invertebrate interactions through disrupted food webs. Invertebrate prey also predominates among forest and farmland birds. The decline of insects is considered to be one of the most important factors in the decline of species living in agricultural landscapes [4]. Our studies revealed fourteen decreasing and five increasing species of insectivores in farmland. However, the overall decline in insectivorous birds illustrates the general patterns of decline in invertebrate populations, diversity, and abundance [106–108]. According to our results, the proportion of species with increasing or stable trends among all invertebrate feeding species ranges from 4% for wetland species to 17% for forest species. This is consistent with other studies that have found a general decline in invertebrate feeding species [109].

According to our results, the long-distance migrants were one of the guilds with the strongest declining population trends. All of them were invertebrate feeding species that occupied various habitats, such as forests, wetlands, and cultural landscapes (Supplementary Material, Table S2). The wintering areas of long-distance migrants are geographically and ecologically separated from their breeding areas, so these populations are affected by a wider range of ecological and environmental factors in breeding and wintering areas or during migration [110]. In addition, low phenotypic plasticity in long-distance migrants prevents birds from adequately adapting to environmental changes [111]. Due to lower flexibility, species adaptation may not be synchronous with current rapid environmental changes [112], which could lead to a mismatch of population processes with natural events that are essential for species survival (i.e., trophic mismatch) [113,114]. On the other hand, the rate of decline of non-migrants was similar to that of long-distance migrants (Table 3), indicating constraints on birds that live predominantly in breeding areas. The nine resident species in our study were non-migrants throughout Europe, which means that the

ringed birds of these species represent local Slovenian breeding populations. The decline of non-migrant populations was found in different habitats, suggesting that the species face unfavourable conditions in various habitats in the country. Similar results on the greater decline of sedentary birds were also obtained in several other studies [97,115,116], indicating a significant influence of ecological and environmental conditions on bird populations in breeding areas [98] throughout Europe and throughout the year.

On the other hand, the short-distance migrant group had one of the least declining trends of all trait groups and included the greater number of species with increasing populations among birds clustered by migration distance (12 species; Supplementary Materials, Table S2). Positive or stable trends in short-distance migrants, also identified in other studies [97,117,118], are likely related to a more flexible phenotype in these species that allows them to adapt earlier to changing environmental conditions [119].

Most studies identified stable or increasing population trends in the forest guild [86,87,97,120], in contrast to our results, which indicate a predominantly declining trend in forest birds. Nevertheless, the smallest declines of all guilds were found in the forest guild in our study. Some forest species, such as the Eurasian Blue Tit (*Cyanistes caeruleus*), may benefit from their broader ecological niches and survive outside forests, such as in parks and gardens. Some other species, such as the Nuthatch (*Sitta europaea*), have been observed to expand their range due to high productivity, which promotes population growth and allows the species to expand [121,122].

Seed eaters represented a group with the least declining trend (Table 3), which was also observed in other studies [97]. Interestingly, this includes a larger number of species that are increasing among farmland seed eaters, such as the Yellowhammer (*Emberiza citrinella*) and the Corn Bunting (*Emberiza calandra*). Their population growth may be related to improved winter herbaceous flora promoted by cropland programmes [123].

## 5. Conclusions

In conclusion, we have observed a long-term and widespread decline of passerine species during autumn migration. Moreover, widespread declines in avifauna across a range of vital and behavioural traits, and across a range of spatial and ecological scales, indicate widespread environmental changes in Europe. Ecosystems altered by humans are leading to an increase in the number of rare bird species that until recently were common. These birds lack the life history traits typical of intrinsically rare species, and are at greater risk of extinction [124]. On the other hand, populations of species with greater productivity and more flexible behaviour, such as short-distance migrants, have the greatest chance of succeeding in the recently rapidly changing environment because they can adapt to changes in a timely manner. Moreover, due to their population growth and dispersal tendency, they might be able to occupy the empty niches of locally extinct specialists, especially species with low productivity, or even competitively displace them [125]. According to our data, recent trends are toward ecosystem homogeneity, with impoverished species diversity and avifauna, including a few species that are increasing in abundance. Accordingly, our conservation efforts should be directed at limiting activities that adversely affect wildlife, especially in natural and semi-natural habitats, and implementing conservation actions in a broader range of habitats to maintain diverse bird communities at different spatial scales. In doing so, eastern European populations, which according to our data have been the most stable, could serve as a basis for recolonisation of other regions in Europe.

**Supplementary Materials:** The following supporting information can be downloaded at: https://www.mdpi.com/article/10.3390/d14110905/s1, Table S1: Average number of annually ringed birds with standard deviation (SD), coefficients of variation, maximal number of locations and number of ringing years. Below the table are listed species that were ringed during autumn migration during the study period (2000–2016), but were due to low presence not included in modelling analyses; Table S2: All species for modelling and their ecological and life-history traits. See abbreviations for traits in Table 1; Table S3: Percentages of recoveries by countries in geographic group after spatial clustering; Figure S1: Models of population dynamics of studied passerine species on autumn

migration in Slovenia according to long-term ringing data; Figure S2: Modelling of population dynamics and trends for particular species guild according to ecological traits: (a) habitat traits, (b) diet traits, and life-history traits: (c) productivity for the period 2000–2016. See Table 1 for abbreviations; Figure S3: Modelling of population dynamics and trends for particular species guild according to: (a,b) migratory behaviour, and (c,d) species occurrence in breeding season in Europe and Slovenia for the period 2000–2016. The model for OB (occasional breeders) is not significant; its explained deviance is <10%. See Table 1 for abbreviations; Figure S4: Modelling of population dynamics and trends for particular species guild according to: breeding origin (clusters of recoveries) for the period 2000–2016. See Table 1 for abbreviations.

**Author Contributions:** Conceptualization, T.P. and A.V.; methodology, T.P. and A.V.; validation, T.P.; formal analysis, T.P.; investigation, T.P. and A.V.; resources, T.P. and A.V.; data curation, A.V.; writing—original draft preparation, T.P.; writing—review and editing, A.V.; visualization, T.P.; supervision, A.V.; funding acquisition, T.P. and A.V. All authors have read and agreed to the published version of the manuscript.

**Funding:** Bird ringing in Slovenia is financially supported by the Ministry of Culture of the Republic of Slovenia. The study is a part of T.P.'s PhD work, which is financially supported by the Ministry of Education, Science and Sport. A.V. was supported by the Ministry of Culture of the Republic of Slovenia and research core funding No. P1-0255 by the Slovenian Research Agency.

**Institutional Review Board Statement:** The bird ringing was licensed by the Slovenian Environment Agency (Ministry of the Environment and Spatial Planning of the Republic of Slovenia) to the Slovenian Museum of Natural History, where Slovenian Bird Ringing Centre operates.

**Data Availability Statement:** In the study we used data from the public Slovenian bird ringing database, which is managed by Slovenian Museum of Natural History. The data can be obtained upon request from Slovenian Bird Ringing Centre (Slovenian Museum of Natural History: https://www.pms-lj.si/si/o-naravi/zivali/vretencarji/ptici/slovenski-center-za-obrockanje-pticev, accessed on 27 September 2022). The data on European trends of common birds was sourced from EBCC/BirdLife/RSPB/CSO. https://www.ebcc.info/trends-of-common-birds-in-europe-2017-update/, accessed on 7 February 2020).

**Acknowledgments:** We are particularly grateful to the volunteer ringers from the Slovenian Bird Ringing Centre (Slovenian Museum of Natural History) who provided data on birds in the period 2000–2016: Dušan Belingar, Franc Bolta, Darjo Bon, Ivo Božič, Franc Bračko, Igor Brajnik, Jože Bricelj, ml., Jože Bricelj, st., Alfonz Colnar, Stane Černalogar, Marjan Debelič, Dušan Dimnik, Jože Dolinšek, Stanko Drašček, Dare Fekonja, Jernej Figelj, Jože Geiser, Marjan Gobec, Jože Gračner, Dejan Grohar, Peter Grošelj, Vojko Havliček, Ludvik Jakopin, Marko Jankovič, Tone Jankovič, Leon Kebe, Milovan Keber, Brane Koren, Stane Kos, Jelko Kozjak, Brane Lapanja, Ivan Lipar, Anton Lisec, Zvonko Lončarevič, Tone Macele, Tomaž Mihelič, Jurij Mikuletič, Jože Nered, Žan Pečar, Miro Perušek, Dušan Petkovšek, Rajko Piciga, Zdravko Podhraški, Dušan Pogačar, Milan Pustoslemšek, Aljaž Rijavec, Miran Romšak, Luka Simončič, Branko Slabanja, Andrej Sovinc, Željko Šalamun, Dare Šere, Iztok Škornik, Pavle Štirn, Vlado Štolfa, Polde Štricelj, Rudolf Tekavčič, Tomi Trilar, Andrej Trontelj, Miro Vamberger, Lojza Vesel, Bogdan Vidic, Milan Vogrin, Iztok Vreš, Davorin Vrhovnik, Eva Vukelič and Ivan Zlobko. Special thanks go to Dare Šere, who coordinated bird ringing in Slovenia from 1987 to 2011, and to Dare Fekonja, who has coordinated bird ringing in Slovenia since 2012. We would like to thank Andrej Kapla for map preparation.

**Conflicts of Interest:** The authors declare no conflict of interest. The funders had no role in the design of the study, in the collection, analyses, or interpretation of data, in the writing of the manuscript, nor in the decision to publish the results.

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
