# Peer review of "Long-Term Ringing Data on Migrating Passerines Reveal Overall Avian Decline in Europe"

_diversity, doi:10.3390/d14110905_

Round 1

Reviewer 1 Report

This paper represents interesting results of the analysis of 17-years ringing data from autumn migration in Slovenia that were used to estimate population trends of 85 passerine species and to identify whether guilds of species with similar movement status, life history traits, common ecological characteristics, and geographic origin share a common trend direction. The results confirms the widespread decline of European passerine birds, indicating widespread environmental changes in Europe.

The paper is well-writen and clear.

I suggest to exclude species with the insufficient sample size from the analysis, even when smoothing process resulted with significant models. If included, their trends are influencing the trends of guilds where they belong and might result with the misinterpretation of results (see ln 304-315).

Some smaller comments are provided:

Tables and Figures (incl supplementary materials) are not numbered following the position in the text. Please, check it throughly and correct where needed.

Ln 82-ln 84 The same route is called „southeastern (Balkans)“ (ln 82) and eastern European (ln 84) that makes confusion; especially because in the ln 83 the eastern (Indian) route is mentioned. Use route names consistently through the manuscript!

ln 83-84; Ln 90-91. Authors claim that Atlantic route passes through Slovenia. It is correct, as some birds (such as ACRSCI) that migrate through Slovenia join the Atlantic route. However, as this route mainly passes north of Alps and north of Slovenia, additional sentence should be add to make it clear to the readers.

ln 96-97  Please, correct the mistake in Figure 1 caption: red dots – NNE, blue dots – NN (now is vice versa)

ln 107 Supplementary material Table S2 is reffered to before S1. Please, correct it.

Ln 112 – Instead of Appendix 1 it should probably be Table S1, But see the previous comment!

Ln 109 & ln 122 – Table 2 is reffered to before Table 1 (ln 153). Please, correct it!

Ln 188-190 – It is not clear how you identified this seven species based on the data in Table S2., as more than seven species has the coefficient of variation among no. of ringed birds < 30%

Ln 240 & ln 252 Figure S4 was refered to before figure S3

Ln 249. There are no bolded interactions in the table 4 (mentioned in the table caption). Please, add bold format where needed.

Ln 279-280 use: do not cover, instead of: might not cover. See the Eurasian African Bird Migration Atlas – European populations of the majority of species included in the analysis use several routes, many of them not passing through Slovenia

Ln 361 use „both“ instead of „all“ for only two species

Ln 434 use rare bird species instead of rare birds for the more clear statement

Author Response

Please see our response in attached file.

Al Vrezec

Reviewer 2 Report

I think this is a nice and interesting paper, that however could be improved by clarifying some methodological procedures that I have found obscure. I explain below which parts of methods should be better explained to improve understanding of the paper.

L 122: I believe more details should be provided on the characteristics of the GLM

L141-143. I would appreciate a more detailed explanation of the process of recoveries clustering. Why the Manhattan distance was used?

L 155-156: “we converted population indices to the same decimal basis before assessing common trends for each group” I believe this needs to be explained more clearly

L 165: please explain how the trends wee classified in categories por ordinal regression analysis. Are the categories Stable, Decline, Icrease? The modeling of trends for each guild category is not clearly explained

L 207-209: explain the meaning of DE

L 240: should be Figure S2 instead of S4?

L 251-255: This result is difficult to interpret

Figure S2. Graphics are too small; thus this figure could be divided in several figures to allow for larger graphics

Figure S3: explain the reason for presenting the red dashed line at 10%

Figure S4. Numbers are sated to correspond to species, but, do they correspond to species id numbers that are shown in Table S1? If so, why are some numbers shown in both graphics? For instance 6, 22, 23 …. How was species importance quantified to determine their colors?

Author Response

Please find our response attached.

Al Vrezec
